# Acquisition of molecular rolling lubrication by self-curling of graphite nanosheet at cryogenic temperature

Panpan Li [1,2,5], Wenhao He [1,5], Pengfei Ju[3], Li Ji [1,2] ✉, Xiaohong Liu[1], Fan Wu[4], Zhibin Lu [1,2] ✉, Hongxuan Li [1,2] ✉, Lei Chen[1,2], Jingzhou Liu[3], Huidi Zhou[1,2] & Jianmin Chen [1,2] ✉

Friction as a fundamental physical phenomenon dominates nature and human civilization, among which the achievement of molecular rolling lubrication is desired to bring another breakthrough, like the macroscale design of wheel. Herein, an edge self-curling nanodeformation phenomenon of graphite nanosheets (GNSs) at cryogenic temperature is found, which is then used to promote the formation of graphite nanorollers in friction process towards molecular rolling lubrication. The observation of parallel nanorollers at the friction interface give the experimental evidence for the occurrence of molecular rolling lubrication, and the graphite exhibits abnormal lubrication performance in vacuum with ultra-low friction and wear at macroscale. The molecular rolling lubrication mechanism is elucidated from the electronic interaction perspective. Experiments and theoretical simulations indicate that the driving force of the self-curling is the uneven atomic shrinkage induced stress, and then the shear force promotes the intact nanoroller formation, while the constraint of atomic vibration decreases the dissipation of driving stress and favors the nanoroller formation therein. It will open up a new pathway for controlling friction at microscale and nanostructural manipulation.

Friction is a common phenomenon that exists ubiquitously in industry and daily life, and yet is the main cause of energy loss and material consumption accompanying wear. It is estimated that friction account for 30% waste of primary energy, while wear causes 60% failure of equipment approximately[1,2]. The history of exploring friction phenomenon is long and the controlling level has improved, which promote the progress of industrial civilization greatly[3–6]. In the macroscale world, the design of wheels and bearings has facilitated the rolling replacing sliding, which reduces friction significantly (by more than 100 times) and lays the foundation for modern industry as well as

convenient transportation, and wheels are therefore known as the most crucial mechanical invention in history. In the microscale world, the layer-layer slipping mechanism of laminar materials with weak interlayer interaction is mainly responsible for the friction reduction nowadays, but it is ineffective under many conditions[7–10], especially in extreme cryogenic and spacial environment. For example, layered graphite with excellent lubricity in air but tends to fail in vacuum[11,12], which has caused the major air crash accidents, and still is a troublesome puzzle in the tribological field. While the molecular rolling lubrication is a more universal and efficient principle[13,14], and if it can be

[1]State Key Laboratory of Solid Lubrication, Lanzhou Institute of Chemical Physics, Chinese Academy of Sciences, Lanzhou, China. [2]Center of Materials Science and Optoelectronics Engineering, University of Chinese Academy of Sciences, Beijing, China. [3]Shanghai Aerospace Equipment Manufacture, Shanghai, China. [4]Changchun Institute of Applied Chemistry, Chinese Academy of Sciences, Changchun, China. [5]These authors contributed equally: Panpan Li, Wenhao He. ✉e-mail: jili@licp.cas.cn; zblu@licp.cas.cn; lihx@licp.cas.cn; chenjm@licp.cas.cn

obtained, another breakthrough for controlling the friction actively will bring undoubtedly[15–17]. However, this hypothesis has been proposed for decades in theory, but it has not been realized and proven in experiment so far[18,19].

Cryogenic temperature is a typical extreme environment, where the effective lubrication still is a challenge and most of lubricating materials perform poor tribological properties at present[20–22]. Key moving parts in lunar project and deep space exploration involve the cryogenic lubrication problems, and the reliable lubricants service still is troublesome[23–26]. On the other hand, many fascinating scientific discoveries are observed at cryogenic temperature[27–30], due to the motion of microparticles (electrons, phonons and atoms etc.) are greatly inhibited, and a lot of unknown scientific effects exist thereat.

This study finds an edge self-curling nanodeformation phenomenon of graphite nanosheets at cryogenic temperature (77 K), which is then used to promote the formation of graphite nanorollers in the friction process towards molecular rolling lubrication (Fig. 1a). Graphite fails in vacuum lubrication in general, and it is a long-standing problem in the tribology field. While herein graphite exhibits abnormal macroscale lubrication performance (frictional coefficient decreases by 5 times and wear rate decreases by 83 times as compared with graphite at 300 K) at cryogenic temperature (50 K) in vacuum, which give a typical demonstration and provide a universal molecular rolling lubrication designing principle to overcome the failure problem in extreme environments of the existing layered lubricant system. Then influencing factors, structural evolution and driving mechanism of the self-curling nanodeformation and molecular rolling lubrication mechanism are investigated systematically by experiments and simulations.

## Results

### Molecular rolling lubrication at cryogenic temperature

The microstructural evolution and tribological behavior of graphite at normal (300 K) and cryogenic temperature (50 K) in vacuum are shown in Figs. 1 and 2. It indicates that layer-layer sliding graphite fails quickly at 300 K, corresponding to the interfacial microstructure of broken graphite fragments and amorphous carbon (Fig. 1b), thus inducing high friction and severe wear (Fig. 2a, b), because defects and edge bonds exhibit strong chemical interaction, which is consistent with other reports[11,31]. While at 50 K, a large amount of ordered nanoscroll crystal lattices are formed at the friction interface (Fig. 1c, Supplementary Figs. 1 and 6). The enlarged view (Fig. 1d) of a nanoscroll suggests that it is formed via rolling of trilayer GNSs from head to tail, and the lattice spacing (0.34 nm) is consistent with the d-spacing of the (002) basal plane of graphite. Corresponding axial views (Fig. 1e, f) of the nanoscroll lattices reveal that they are nanoroller structure with the length of hundreds of nanometers, and the lattice spacing is consistent with (100) crystal plane spacing of graphite. Furthermore, the scanning electron microscope (SEM) morphology (Fig. 1g) of the friction interface shows that nanoscrolls are distributed parallely rather than disorderly, which suggests the nanoscrolls are indeed nanorollers and function as molecular bearings to perform rolling rather than sliding during the friction process. This lubrication effect is also confirmed by the awaken lubricity of graphite at 50 K (Fig. 2a, c), where the friction coefficient is low and stable (0.04–0.06), and the wear track is narrow and shallow. Then wear rate of graphite at 50 K is only about 1/83 of that at 300 K (Fig. 2b, c, Supplementary Fig. 2). The dynamic friction coefficient at alternating temperatures (300 K-50 K-300 K-50 K; Fig. 2d) decreases as the temperature declines, and stabilizes at a minimum (about 0.05) when the

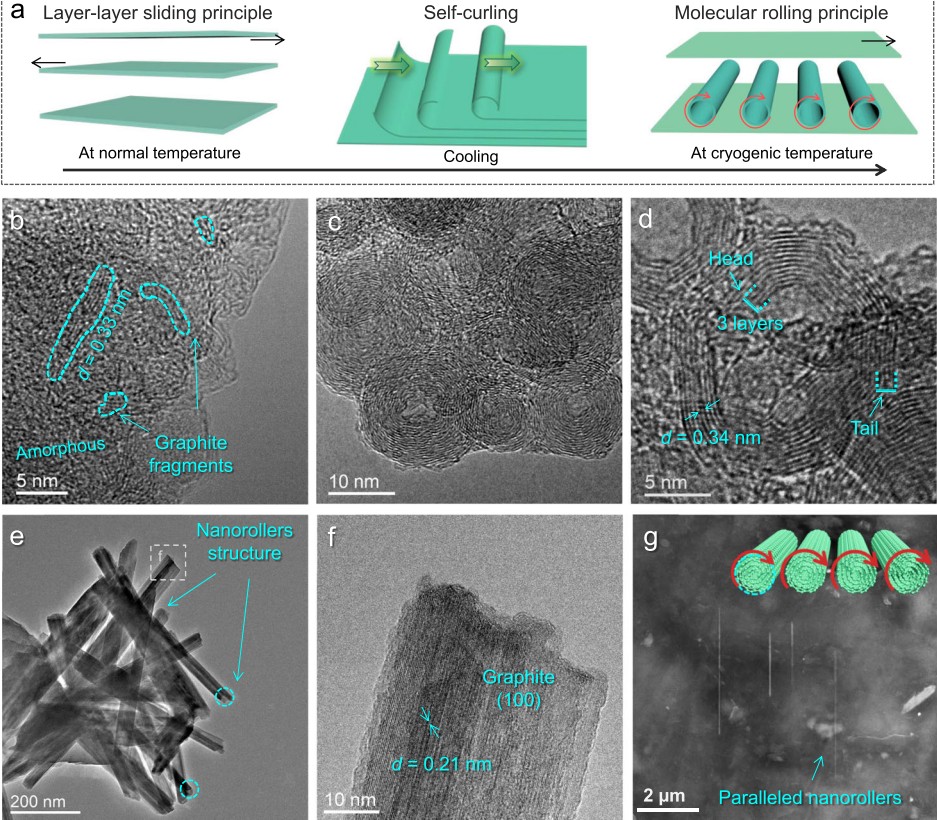

**Fig. 1 | The illustration of the molecular rolling lubrication of graphite at cryogenic temperature. a** Schematic of layer-layer sliding and molecular rolling lubrication as well as the structural evolution. **b** Microstructure of the friction interface at 300 K. **c** Microstructure of the friction interface at 50 K. **d** The enlarged view of the formed nanoscroll. **e, f** TEM images of the axial view of the formed graphite nanoroller. **g** SEM morphology of the friction interface at 50 K.

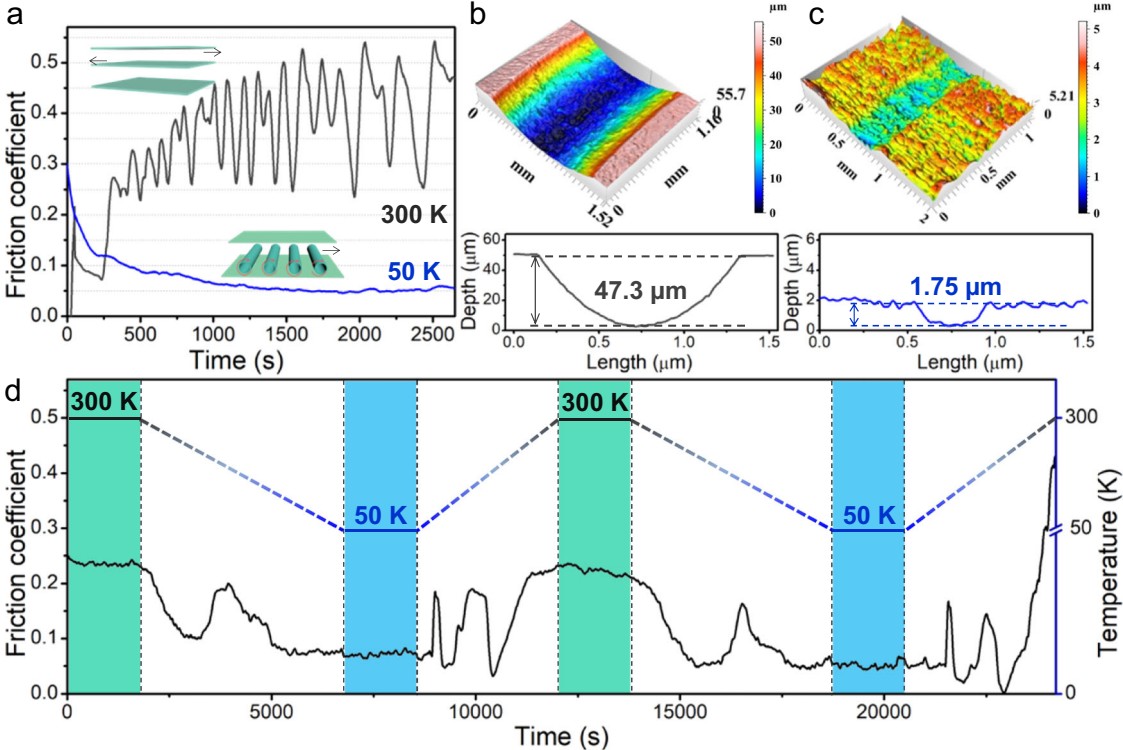

**Fig. 2 | Friction and wear behaviors of graphite at 300 K and 50 K in vacuum.** **a** Friction coefficient curves of graphite at normal of 300 K and cryogenic temperature of 50 K in vacuum. **b** Three-dimensional (3D) profile and wear scar depth after friction at 300 K. **c** 3D profile and wear scar depth after friction at 50 K, and **d** friction coefficient curve of graphite under alternating temperatures (300 K and 50 K) in vacuum.

temperature is lower than 100 K. Both the static and dynamic friction tests demonstrate that the cryogenic tribological performance of graphite in vacuum is preternatural in traditional tribological research[6,32,33]. The hypothesis of molecular rolling lubrication as wheel is a long-term dream of many tribologists, but unfortunately, it has not been proven experimentally for decades[15,18,19]. This work provides the first conclusive experimental evidence for the occurrence of molecular rolling lubrication and its contribution to macroscale low friction, which will open up a new pathway for controlling molecular-scale rolling friction, meaningful for obtaining new comprehension and countermeasure for graphite vacuum lubricating failure, a long-standing problem in the tribology.

### The self-curling nanodeformation phenomenon of GNSs at cryogenic temperature

To reveal the formation mechanism of molecular nanorollers at cryogenic temperature, influencing factors, including the graphite defect characteristics, interfacial interaction, and static cryogenic temperature are investigated experimentally (Supplementary Figs. 3–13). Although the fewer defects and weaker interfacial interaction are favorable for the formation of nanorollers, neither of them is the decisive one but cryogenic temperature. Then, the impact of temperature on the structural changes of GNSs is explored by in-situ cooling TEM observation (Fig. 3a, b, Supplementary Fig. 14). As the temperature varies from 300 K to 77 K (liquid nitrogen temperature), the overlapping GNSs become separate and the horizontal (001) crystal plane transforms into the vertical (002) basal plane, indicating cryogenic temperature leads to self-curling of the GNSs edges, the inserts show diagram. Then after the friction process, the curled GNSs are transformed to intact nanorollers, and high-resolution transmission electron microscopy (HRTEM) images show the structural and morphological evolution of the nanoroller formation experimentally (Fig. 3c–e). Supplementary Fig. 15 demonstrates the structural

evolution of the nanoroller formation during the dynamic friction process, including nanosheet-delaminating, self-curling and shear-rolling three stages.

Molecular dynamics (MD) simulations are conducted to disclose the experimental phenomenon abovementioned. Under the uniform temperature field, the GNSs cannot spontaneously curl at any temperature. While there is a temperature difference between the upper and lower surfaces, the GNSs can be induced to undergo self-curling. In fact, uneven temperature distribution inevitably exists during the cooling and friction process. For instance, the lower surface of the sample is cooled and the upper surface is exposed to normal temperature during the low-temperature friction and cooling process. Viewing this fact, we simulate and calculate the structure and stress changes when the temperature of one side of the bilayer GNS is lowered from 300 K to 70 K (Fig. 3f, i). The atomic spacing becomes smaller on the cold side (in-situ cryogenic XRD; Supplementary Fig. 16), which causes lattice distortion and stress (Fig. 3h). Thus, then opposite stress is generated at the cold and hot sides (Fig. 3i), and drives GNSs to curl. Along with the stabilization of temperature and the edge-curling of GNSs (as the relaxation time is prolonged), the stress of GNSs is gradually released to reach an equilibrium state. The friction force during the friction process further drives the edge-curled GNS rolling to form the intact nanoroller (Fig. 3g). Experiments and simulations show the consistent results of the structural and morphological evolution of GNSs during the rolling process. During the graphite nanorollers formation process, the potential energy of the GNSs/nanoroller/graphite nanosheet system continually decreases (Fig. 3j), which indicates that the formed graphite nanorollers are energetic favorable. In a word, it suggests that the driving force of the self-curling is essentially from the stress induced by the temperature gradient and uneven atomic shrinkage during the cooling stage, and then the shear force further drives it to a more stable energy state of intact nanorollers.

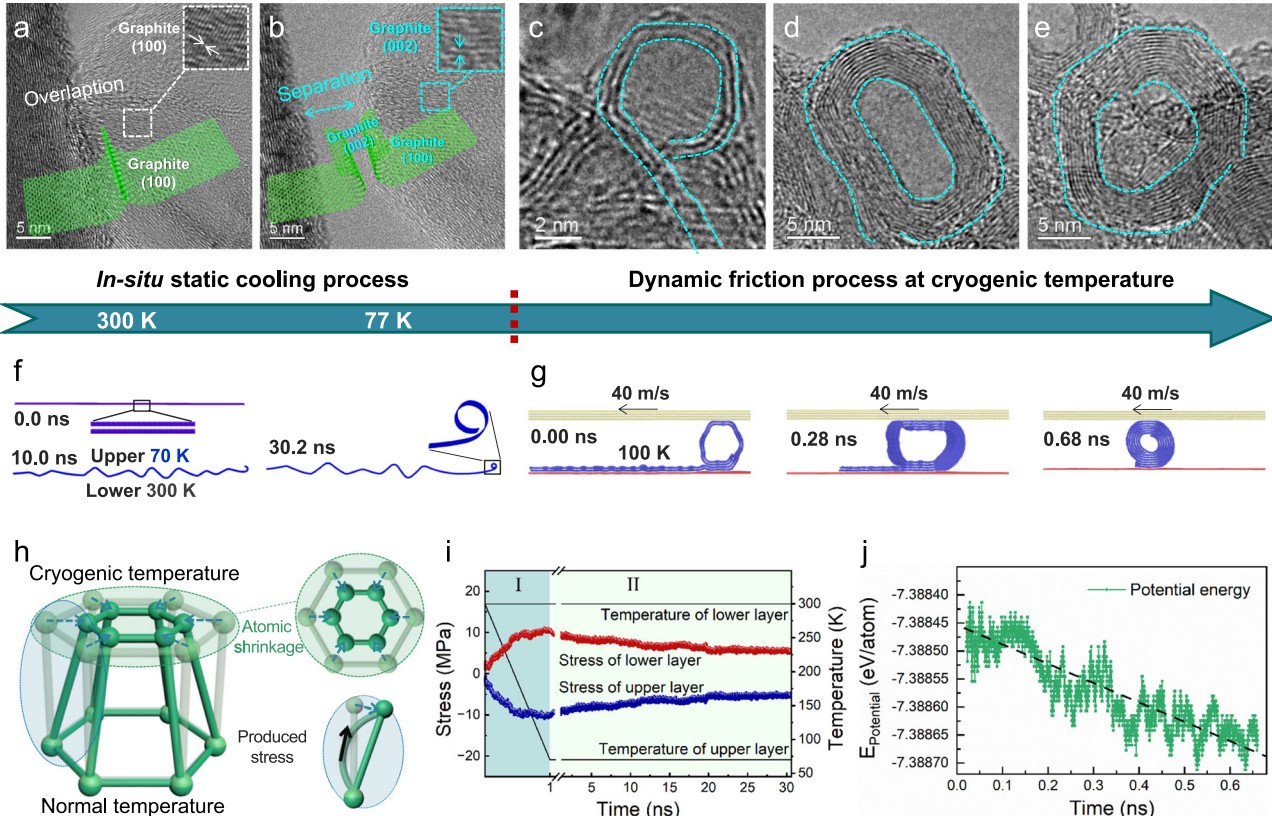

**Fig. 3 | MD simulation and experiment of the formation and mechanism of graphite nanoroller. a, b** In-situ HRTEM images observation and schematic diagrams of GNSs from 300 K to 77 K. **c–e** HRTEM images of the structural evolution of graphite nanoroller during the friction process experimentally. **f** Simulated formation of the self-curling of GNSs under temperature gradient from 300 K to 70 K. **g** Simulated formation of the intact graphite nanoroller during the friction process at 100 K. **h** Schematic diagram of stress generation induced by lattice distortion of atomic shrinkage. **i** Changes of stress of upper and lower layers of GNSs along x-direction with time (stage I corresponds to the temperature dropping of the atoms in the upper layer from 300 K to 70 K, and the temperature keeps constant at stage II). **j** Potential energy curves of GNSs/nanoroller/graphite nanosheet system after the formation of nanorollers. The black dotted lines are shown to guide the eyes.

Considering the graphite has a low thermal conductivity along the out-of-plane direction, and the internal stress may have a large difference for the GNSs with different layers. The effect of number of layers on self-curling is further investigated by MD simulation (Supplementary Figs. 17 and 18). The stress to curl becomes greater with increasing the number of layers, and that only with 1–3 layers can occur self-curling. This is in consistent with the extensive HRTEM results, where the rolling unit can be distinguished from the head or tail.

### The effect of atomic vibration on self-curling nanodeformation phenomenon

Then influence of temperature on the self-curling nanodeformation and molecular nanorollers formation is investigated. As the dynamic friction test at alternating temperatures (Fig. 2d) demonstrates that only when the temperature reaches a certain limit low level (100 K) can a special stable and low lubricating state be achieved. It means that applying cryogenic temperature to generate the temperature difference is the prerequisite for inducing the stress-driven rolling process of GNSs in vacuum. Then in-situ cryogenic Raman spectra and friction interface microstructure analysis are performed to investigate whether cryogenic temperature has other effects on the nanodeformation of GNSs in vacuum. A nonlinear drop of the Raman signal intensity appears at 100 K in comparison with the one at 110 K (Fig. 4a), which indicates that the atomic vibration[34–36] is greatly inhibited at 100 K, since the Raman intensity is proportional to the atomic vibration[37,38]. Meanwhile, TEM images of friction interfaces show that 100 K is the critical temperature for the nanorollers formation (Fig. 4b). This indicates that the suppression of atomic vibration at cryogenic

temperature (such as 100 K) favors the formation of nanorollers. As the typical 2D materials with laminar structure, GNSs exhibit strong in-plane constraint with weak atomic vibration but weak out-of-plane constraint with strong atomic vibration, and the out-of-plane vibration amplitude at normal temperature even can exceed the layered thickness[39,40]. Thus the influence of atomic vibration will appear in the force (mechanical force, shear force etc.) bending and nanodeformation process, which indicates atomic vibration would dissipate a large number of driving stress (or energy) in the out-of-plane curling deformation process and obstruct the curling deformation. Based on this recognition, to bridge the connection between the driving stress of curling deformation and atomic vibration during the nanoroller formation process, we simulate the atomic stress distribution of GNSs during the self-curling process under temperature gradients from 70 K to 300 K and from 300 K to 570 K. Simulations indicate that the distribution of atomic stress is dispersed at 300 K, while it becomes concentrated at 70 K (Fig. 4d, e). It demonstrates that strong atomic vibration indeed dissipates a large part of driving stress (or energy) and hinders the curling of the GNSs. The weak atomic vibration at cryogenic temperature is conducive to stress transmission, thereby reducing energy dissipation, and thus is easier to perform mechanical deformation at cryogenic temperature. We believe that it can enhance the cognition of the force law of microparticles and the active control of the molecular configuration manipulation in solid.

### Mechanism of molecular rolling lubrication

The force on objects at microscale is quite distinct from that at macroscale owing to the size effect[4,41]. Density functional theory

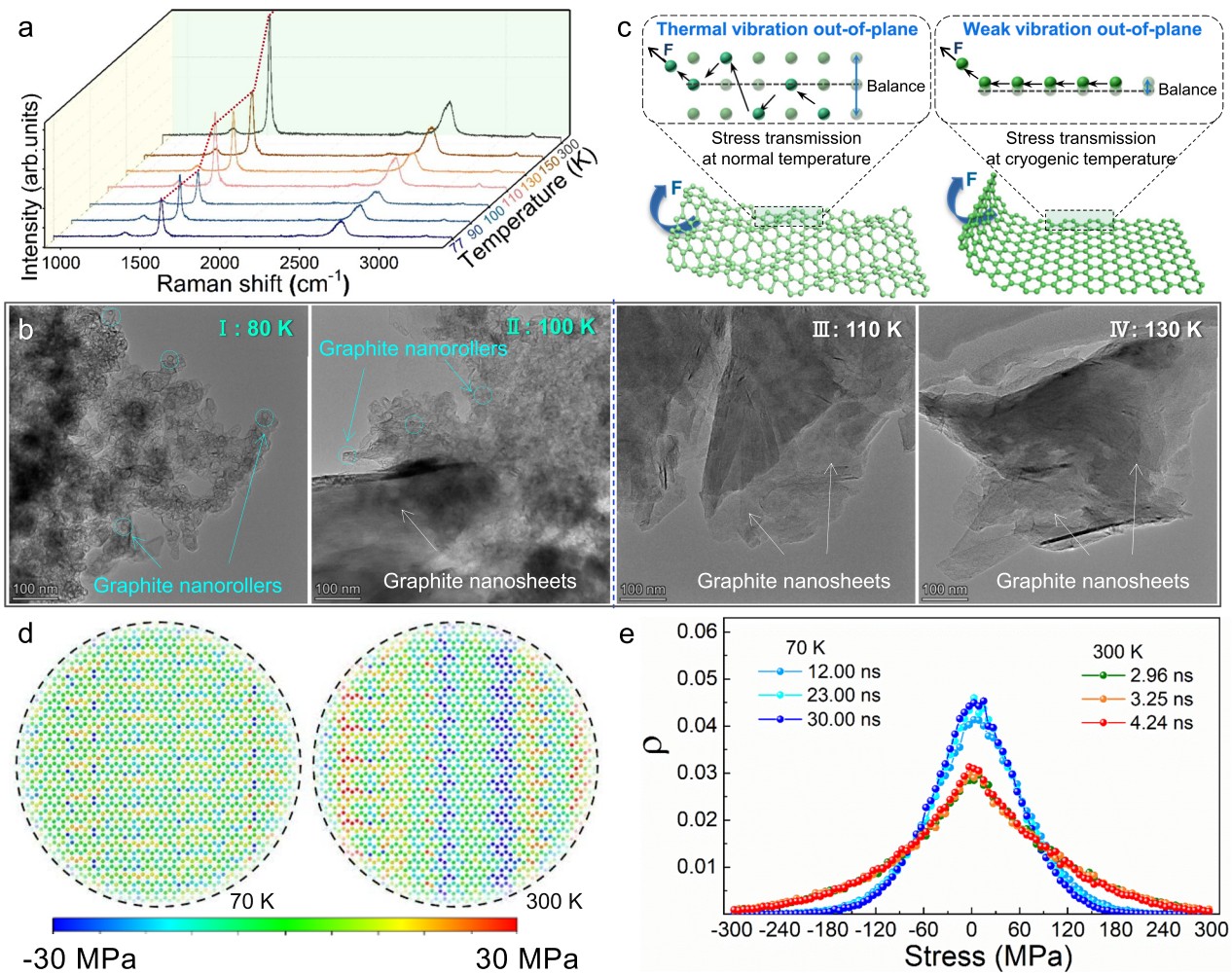

**Fig. 4 | The effect of atomic vibration on the curling of GNSs at normal and cryogenic temperatures. a** In-situ Raman intensity variation upon varying temperature. **b** Interfacial microstructure of graphite at different temperatures friction process. **c** Schematic diagrams of the stress transmission at normal and cryogenic temperatures. **d, e** Comparison of the atomic stress distributions at different temperatures during MD simulations. Quantitative statistics of atomic stress distribution are obtained by counting all atoms in the nanosheets with 70 K and 300 K, respectively, and the two systems possess the same driving stress. Qualitative atomic stress distribution is shown by selecting partial atoms in one of nanosheet (at 12.00 ns and 70–300 K system as well as 2.96 ns and 300–570 K system).

(DFT) simulation[42] is used to explore the molecular rolling lubrication mechanism at microscale. Figure 5a–f shows the occurrence of carbon nanotube rolling during quasi-static friction. The maximum energy dissipation in the rolling process (5.1 meV) is considerably lower than that in the sliding process (146.4 meV) under the same load of 7.9 GPa (Fig. 5g), and even lower than that in the sliding process under 0 GPa (7.9 meV). Comparison of the charge density distributions at friction interfaces during sliding and rolling reveals that a charge redistribution on the friction interface during sliding, while the charge distribution remains almost unchanged during rolling. That is, the energy dissipation of the rolling is almost two orders of magnitude lower than that of sliding. The lower energy dissipation can be attributed to the negligible change in the differential charge density during rolling (Fig. 5h–k). It indicates that a more efficient rolling lubrication can be achieved by forming graphite nanorollers functioned as molecular bearings, which is supported by both of the experiment and simulation.

## Discussion

In this study, the edge self-curling nanodeformation phenomenon of GNSs at cryogenic temperature is found, which is used to promote the transformation of graphite nanosheets to nanorollers and then achieve the experimental acquisition of the molecular rolling lubrication. To the best of our knowledge, the parallel nanorollers formed at friction interface provide the conclusive experimental evidence of molecular rolling lubrication, and the DFT simulation reveals the rolling lubrication mechanism from the electronic interaction perspective. Graphite awakes abnormal lubrication performances in vacuum with ultra-low friction and wear at cryogenic temperature, reducing friction coefficient from 0.25–0.45 to 0.04–0.05 and solving the long-term problem of graphite lubrication failure in vacuum. Then influencing factors, structural evolution and formation mechanism of the edge self-curling nanodeformation is investigated systematically by experiments and simulations. It demonstrates that the self-curling nanodeformation is driven by the uneven atomic shrinkage induced stress, and then shear force further facilitates the intact nanoroller formation, while the constraint of atomic vibration decreases the dissipation of driving stress and favors the nanoroller formation therein. It will draw the light on active regulation of molecular structure to achieve adjustable performance and applications just by adjusting the temperature, which is important in extensive practical applications of sensing, optics, electricity, mechanics and tribology. This work will help us inspire new perspectives on various fields, including the control of friction and the nanostructural manipulation.

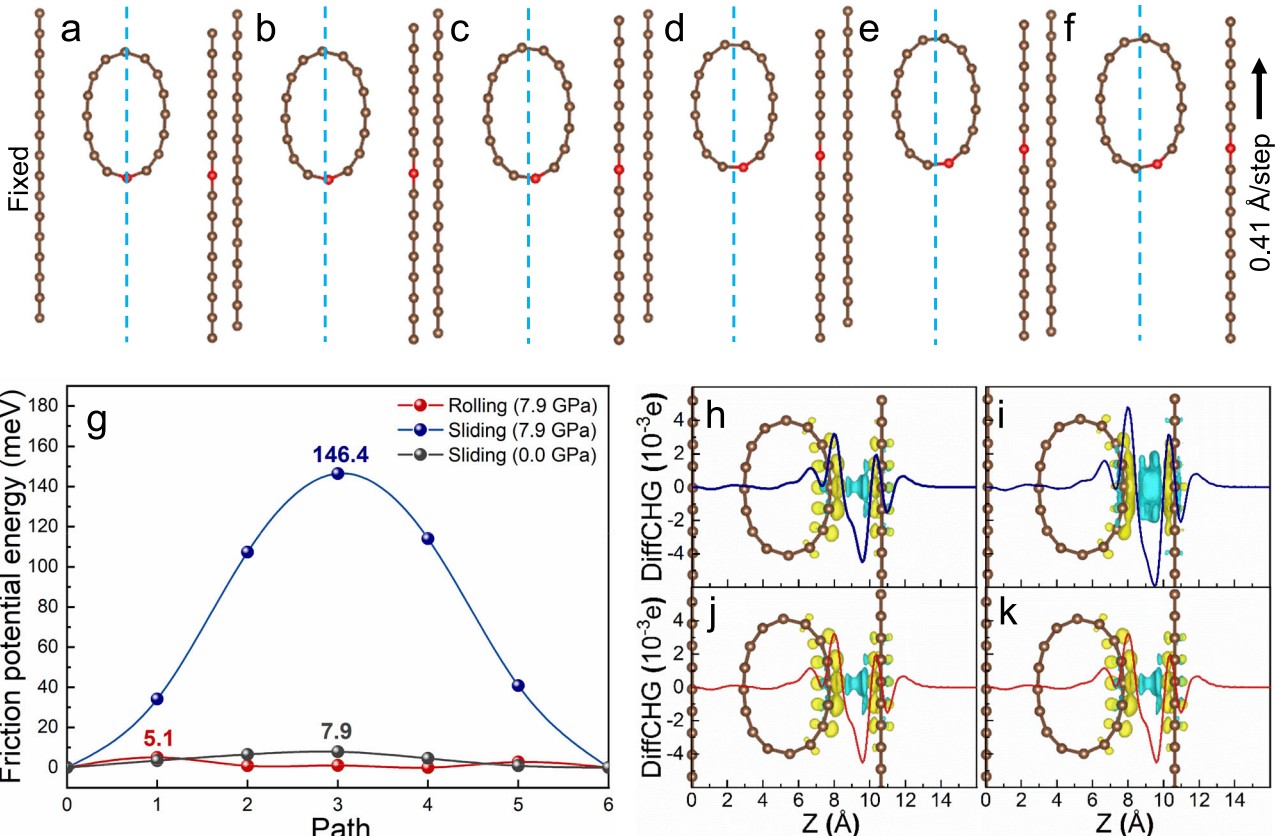

**Fig. 5 | DFT simulation of the molecular rolling lubrication mechanism of graphite nanoroller during the friction process. a–f** The rolling process (the graphite nanoroller is equivalent to a single carbon nanotube to simplify the mode, and details are descripted in Supplementary Fig. 19); the model contains a fixed graphite nanosheet layer, a carbon nanotube allowed to relax during friction, and a graphite nanosheet layer with a velocity of 0.41 Å/step; the diameter of the carbon nanotube is about 6.33 Å and the distance between the two graphite nanosheets is 10.45 Å, corresponding load is 7.9 GPa; the two C atoms are colored with red and the center line of the two graphite nanosheets are shown clearly to reveal the rolling process. **g** The friction potential energy during sliding and rolling processes. The differential charge density and its integral in the plane under 7.9 GPa at sites with the lowest (**h**) and highest (**i**) energy during sliding and rolling (**j**, **k**). The iso-surfaces are $\pm 2.02 \times 10^{-3}$ e/Å³; the yellow and blue refer to charge accumulation and dissipation, respectively.

## Methods

### Materials
Graphite samples were produced with graphite target (product code: 1.1 C.4 N.017, product number: 16-1506) purchased from Dongguan Oulai Sputtering Target Company Limited (Dongguan, China); and the graphite target was prepared through cold pressing of graphite powder (99.99%). Pyrolytic graphite samples were purchased from the Institute of Metal Research of Chinese Academy of Sciences (Shenyang, China), and the pyrolytic graphite target also was prepared through cold pressing. GCr15 bearing steel ball (AISI52100; main component Fe, $\Phi$ = 6 mm, $Ra$ = 10–20 nm) and $Al_2O_3$ ($\Phi$ = 6 mm, $Ra$: 10–20 nm) were purchased from Shanghai Steel Ball Factory Company Limited (Shanghai, China).

### Friction and wear tests
The friction and wear tests were performed with the low-temperature ball-on-disk tribometer (CDW-400, Huayu Aerospace Technology Application Company Limited, Lanzhou). The force sensor (DR-2112, LORENZ, Germany) was used to measure the friction and its accuracy is 0.1 mN to ensure the accuracy of the measured friction force value. The chamber of the tribometer can be manipulated to accommodate vacuum environment, and the temperature can be adjusted from 40 K to 315 K; the schematic diagram of the tribometer is shown in Supplementary Fig. 20, where the graphite (or pyrolytic graphite) disk was driven to slide against the counterpart ball (GCr15 or $Al_2O_3$) in rotary mode. Two thermocouple sensors are used for temperature control,

one is installed near to the sample to measure the real-time temperature and another is installed at the position in the low temperature region. The chamber is cooled by liquid helium; and the entire temperature control system forms a complete loop. When the temperature of the two thermocouples both reached the target temperature in vacuum, the friction tests were started. Prior to the friction test at low temperature (50 K), the vacuum in the chamber was reduced to $5 \times 10^{-3}$ Pa and then the temperature control system was turned on until the temperature of the sample reached 50 K ($<1 \times 10^{-4}$ Pa). The friction test at normal temperature of 300 K ($<1 \times 10^{-3}$ Pa) was performed under the same conditions while the temperature control system was turned off. The friction test was conducted at a rotary speed of 120 rpm and the rotation radius of 5 mm as well as an applied load of 1 N. The dynamic friction test under alternating temperatures (300 K and 50 K) is conducted in the same manners while the target temperature was held at 300 K or 50 K for 30 minutes during friction process. The friction test was continuous except for temperature alternation (heating or cooling), and the friction coefficient was recorded during the whole process. All the friction tests reported in this work were repeated at least three times. The samples used for microstructure characterization of the friction interface were taken at the steady state of friction.

### Characterizations
The phase compositions of the original graphite samples were analyzed by X-ray diffraction (XRD, Bruker, D8 Discover25, Germany; Cu-

$K\alpha$ line at 0.154 nm; tube voltage: 40 kV, tube current: 40 mA). The Raman spectra (Renishaw, Raman inVia, England) were identified at a wavelength of 532 nm (2.3 eV). The samples used for the microstructure characterization of the friction interfaces at the steady state of friction were transferred to the Cu grid, and their morphology was observed by transmission electron microscopy (TEM, FEI Tecnai G2, S-TWINF20, USA; F30 2100 F, JEM, Japan) and HRTEM. The wear scars and wear tracks of the graphite after reaching stable stage of friction were cut with focused ion beam (FIB; FEI, Helios 600, USA) to obtain the cross-sectional specimens. Before the process of Ga⁺ cutting, an Au/Pt layer was deposited onto the surface of the cross-sectional specimens to prevent possible damage, and then their microstructure was analyzed by TEM and HRTEM. The FIB process involves three steps. Firstly, 30 kV of operational voltage and 2.8 nA of operational current were applied to pre-cut the sample. Secondly, 30 kV of operational voltage and 90 pA of operational current were applied to thin the sample to be about 100 nm. Thirdly, the specimens were further thinned to about 70 nm under an operational voltage of 5 kV and an operational current of 17 pA. The wear scars and wear tracks of graphite were observed with an optical microscope (Olympus, STM6, Japan). The morphologies of original surface and wear tracks of graphite (and pyrolytic graphite) were observed with a field-emission environmental scanning electron microscope (ESEM, QUANTA FEG 650, FEI; FESEM, Hitachi, SU8020, Japan) at the accelerating voltage of 30 kV. The 2D and 3D profiles of the wear tracks of graphite after friction at 300 K and 50 K in vacuum were measured with a MicroXAM-800 3D surface profiler (KLA-Tencor, USA). In-situ cryogenic transmission electron microscopy (cryo-TEM, JEOL, JEM F200, Japan) was used to detect the morphological changes of GNSs, and the vacuum chamber of the cryo-TEM can be cooled by injecting liquid nitrogen (77 K) through the sample holder (Gatan 698). For eliminating possible damage of the electron beam to the GNSs, the test current was maintained at the order of several pA during the entire observation process, and the observation was completed as soon as possible in order to eliminate possible impact of the electron beam. Firstly, conventional TEM and HRTEM observations were performed at normal temperature (300 K), and then the electron beam was turned off to get rid of its impact on the test process. The observational field was shifted and located at another GNS. Secondly, the liquid nitrogen was added into the sample holder to cool the vacuum chamber, and the temperature of the chamber was monitored until it reached and stabilized. Finally, return the observational field to the abovementioned target GNSs, and the TEM and HRTEM images at cryogenic-temperature were recorded. In-situ cryogenic Raman (Renishaw, Raman inVia, England) was used to analyze the microstructure change of GNSs during temperature varying process, and the Raman spectra were recorded at a wavelength of 532 nm (2.3 eV). The Raman measurement was first performed at normal temperature (300 K), and then the sample cell was cooled to the target temperature by liquid nitrogen. The temperature was recorded during the cooling process, and the characterization was conducted at pre-set temperatures (150 K, 130 K, 110 K, 100 K, 90 K, and 77 K) when the target temperature was reached and maintained for about 5 minutes. The change of the GNSs structure during temperature-decreasing process was measured to explore the influence of temperature on the graphite structure. In-situ cryogenic XRD (D8 Advance Davinci, Bruker, Germany) was used to identify the crystal lattice change of GNSs during temperature varying process. Firstly, conventional XRD measurement was performed at normal temperature (300 K), and then liquid nitrogen gas was introduced to cool the sample cell. The in-situ cryogenic XRD characterizations were performed at the pre-set temperature (275 K, 225 K, 175 K, 125 K, and 77 K) which was maintained at the target value for about 5 minutes. The phase composition change of the GNSs during the temperature-decreasing process was measured to investigate the influence of the cryogenic temperature on the graphite crystal lattice.

## Model and set-up for DFT calculations

The Vienna Ab initio Simulation Package (VASP)[43] based on the DFT[44] was employed to calculate the total energy of the systems. The projected augmented wave (PAW)[45] pseudopotentials were employed in all calculations within the generalized gradient approximation (GGA)[46,47] with the Perdew-Burke-Ernzerhof (PBE)[48] functional for the exchange and correlation energies. The plane-wave cut-off energy was set as 400 eV. The van der Waals corrections were applied through the Grimme's DFT-D3 method with Becke–Jonson (BJ)[49,50] damping. The Brillouin zone was sampled using a 1*4*1 k-point Monkhorst-Pack definition grid. The systems used to simulate the sliding and rolling are schematically shown in Supplementary Fig. 19. The systems contain a fixed graphite nanosheet, a carbon nanotube with the diameter of about 6.33 Å, a solid graphite nanosheet with a velocity of 0.41 Å/step, and a vacuum layer of 15.00 Å vertical to the layers to eliminate the effects of lattice periodicity. The distance between the two graphite nanosheets is 10.45 Å, corresponding press is 7.9 GPa. For simulating the frictional process, the topmost atomic layer of the systems is moved by 6 steps along a direction. The carbon nanotube was allowed to relax during sliding. For simulating the sliding process, the carbon nanotube was forced to be fixed.

The charge density difference $\rho_{diff}$ (x, y, z) was defined as:

$$\rho_{diff}(x,y,z) = \rho_{total}(x,y,z) - \rho_{up}(x,y,z) - \rho_{down}(x,y,z) \tag{1}$$

where $\rho_{total}$ (x,y,z), $\rho_{up}$ (x,y,z) and $\rho_{down}$ (x,y,z) represent the charge density of the interface system, its upper part and lower part, respectively.

The planar integral $\rho_{(diff\_plane)}$ of the charge density difference $\rho_{diff}$ (x,y,z) is defined as:

$$\rho_{diff\_plane} = \int\int \rho_{diff}(x,y,z)dxdy \tag{2}$$

## Model and set-up for MD simulations

The whole rolling process of GNSs was simulated in two steps. The first step was the formation of initial graphite nanoroller under the temperature gradient field. Supplementary Fig. 21a shows the initial side view of the simulation system containing 160,000 atoms with bilayer graphite nanosheet, and the sizes in a, b and c directions are about 24,600.0 Å, 8.52 Å and 50.0 Å. The atoms about 3.0 Å away from the left edge are fixed during the relaxation. Temperature was controlled with a Langevin thermostat, and periodic boundary conditions were applied in b direction. The simulation process is carried out according to the following steps. Firstly, the system was equilibrated at 300 K for 10 ps, followed by cooling of the upper part from 300 K to 70 K in 1 ns; and then the system was further relaxed until the stress was completely released under the temperature gradient field. Secondly, the initial nanoroller further rolled during friction process; and Supplementary Fig. 21b shows the initial side view of the simulation system. The system contains 113480 atoms, and it is constructed with graphite containing tetralayer GNS, bilayer graphite nanosheet with initial nanoroller and a monolayer graphite nanosheet. The sizes of the system in a, b and c directions are about 393.6 Å, 119.3 Å and 150.0 Å, respectively. A 50.0 Å vacuum nanosheet was added in the c direction and the periodic boundary conditions were applied in three directions. The system was firstly relaxed for 20 ps at 100 K; then a velocity of 0.4 Å/ps was given to the graphite nanasheets to simulate the friction process.

## Potential for MD simulations

The MD simulations were performed by the LAMMPS[51] software package. Adaptive intermolecular reactive empirical bond-order (AIR-EBO) potential[52] was used to carry out all the simulations. The

interactions between the two GNSs were described with the Lennard-Jones (LJ) potential[53] of equational formulation, in which $r$ is the distance between the two GNSs; and the energy parameter ($\varepsilon = 0.00286$ eV) and distance parameter ($\sigma = 3.468$ nm) were used to define the C-C interaction parameters. The open visualization tool (OVITO)[54] software package was employed to show all atomic configurations.

## Data availability

The data supporting the findings of this study are available within the main text and supplementary information files. All data are available from the corresponding author upon request.

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

## Acknowledgements
The authors are grateful to the Strategic Priority Research Program of the Chinese Academy of Sciences (No. XDB0470200), National Natural Science Foundation of China (No. 52275222), Postdoctoral Fellowship Program of China Postdoctoral Science Foundation (No. GZB20230779), West Light Foundation of the Chinese Academy of Sciences (No. xbzg-zdsys-202305), Youth Innovation Promotion Association of the Chinese Academy of Sciences (No. Y202084), Program of Lanzhou Institute of Chemical Physics of Chinese Academy of Sciences (No. KJZLZD-3) and Longyuan Youth Talent Project for financial support. The help in the Raman data collection by Song H. and Zhang Y. P.; and language polishing by Yu L.G., Ma S.H. and Zhao K. is greatly appreciated.

## Author contributions
Li, P. P. and He, W. H. contribute equally to this work. Ji, L., Lu, Z. B., Li, H. X. and Chen, J. M. proposed the project. Ji, L., Li, H. X. and Li, P. P. designed the experiments, and Li, P. P. performed the experiments and analyzed the data under the supervision of Ji, L., Li, H. X. and Chen, J. M.; Ju, P. F., Liu, J. Z. and Liu, X. H. assisted in tribological measurements; Chen, L. and Zhou, H. D. helped to select the sample and discussed the data analyses; Wu, F. assisted in the in-situ cryogenic TEM experiment and TEM data analysis. He, W. H. and Lu, Z. B. conducted the simulation studies and related analysis. Ji, L. drafted the manuscript; Li, P. P. and He, W. H. wrote the manuscript; Ji, L. and Lu, Z. B. helped in organize and revise the manuscript. All authors participated in the data disscussion and the writing of this manuscript.

## Competing interests
The authors declare no competing interests.
