## [Peer Review File · Nature Communications]

Acquisition of molecular rolling lubrication by self-curling of graphite nanosheet at cryogenic temperatureREVIEWER COMMENTS

Reviewer #1 (Remarks to the Author):

The authors observed the spontaneous curling of edges in graphite nano-sheet as the sample is cooled to cryogenic temperature. The self-curling takes an important role in promoting the formation of nano-rollers in the friction tests. The simulation results revealed that the curling of edges is induced by the internal stress in different layers. Compared to the sliding between the flat graphite sheets, a much lower friction coefficient was obtained as the friction was performed on the nano-rollers. The paper presents a new avenue to achieve low-friction at low temperature with the aid of formation of graphite nano-rollers. However, there are some fundamental issues that need to be addressed before considering for publication.

a) After the first cycle of friction test from 300K to 50K, the graphite nano-rollers have been already formed. Why does the friction coefficient become larger again in the following test at 300K?

b) It is still not so clear how the nano-rollers are formed in friction experimentally.

c) The authors emphasize the important role of nano-rollers in reducing friction coefficient. How about simply introducing carbon nanotubes at the frictional interface? Could this produce a similar result?

d) As the temperature decreases from 300K to 77K, self-curling of edges of graphite occurs. The curling is thought to be closely related to the formation of nano-rollers in the subsequent friction. Before friction tests, does the curling still exist when keeping temperature at 77K?

e) The authors accused the formation of nano-roller to be the internal stress induced by the temperature gradient. However, the stress is estimated to be only on the order of 10MPa (Fig. 3i). How does such a small stress cause an obvious curling at edges?

f) The graphite has a very low thermal conductivity along the out-of-plane direction. Even the rolling is induced by temperature gradient, the internal stress may have a large difference in two or three layers. Only a small number of layers could be affected. Here the nano-rollers formed in experiments usually have a large thickness with the layer number more than 10. Could the authors provide more discussion on it?

g) It is proposed the possible contribution of phonons on the formation of nano-rollers. The authors further extend this to the so-called macroscale quantum effect. If so, this may cause more comments on it. For example, how the coupling between phonon and electron on the roller formation?

Reviewer #2 (Remarks to the Author):

Whether rolling friction of nanoparticles plays a role in decreasing friction has long been controversial in the area of tribology. Here by combining experimental and theoretical investigations, the article reported an interesting self-curling of graphite nanosheet at cryogenic temperature. Especially, the authors provide direct observation of the paralleled nanoroller structure on the frictional surface, and hence convince the important role of rolling friction in the system. On the other hand, it was generally accepted that friction of materials will increase with decreasing temperature due to thermal activation effect. However, the authors reported an unusual behavior for graphite nanosheet showing ultralow friction at cryogenic temperature, which is of potential use in industrial applications. I recommend acceptance of the article after some minor points are addressed.

1. Full name for abbreviation GNS should be given. Page 3 Line 66, CNS should be GNS?

2. It seems that Figure 1c and Figure 1d were not ordered according to text.

3. In molecular dynamics simulation, during the cooling of the cold side from 300 K to 70 K for 1 ns

period, how does the temperature of the hot side change? Please clarify.

4. The nanoroller structure (cylindrical shape) should be different from onion-like structure, which by name is of spherical shape. Please correct the use of onion-like in the text.

5. Page 8 Line 170, 4f should be 4e.

Response to the reviewers' comments

Manuscript ID: NCOMMS-23-61566

First, we would like to thank your works and the detailed comments to improve our manuscript. They are meaningful and useful to revise our manuscript to be a more qualified one.

Please check the word file of **Revised Manuscript** and **Revised Manuscript 2** with the track change mode. The revised parts of the manuscript were as following and marked in **red color** in the Revised manuscript 2.

Reviewer #1 (Remarks to the Author):

The authors observed the spontaneous curling of edges in graphite nano-sheet as the sample is cooled to cryogenic temperature. The self-curling takes an important role in promoting the formation of nano-rollers in the friction tests. The simulation results revealed that the curling of edges is induced by the internal stress in different layers. Compared to the sliding between the flat graphite sheets, a much lower friction coefficient was obtained as the friction was performed on the nano-rollers. The paper presents a new avenue to achieve low-friction at low temperature with the aid of formation of graphite nano-rollers. However, there are some fundamental issues that need to be addressed before considering for publication.

Response:

We are grateful to you for taking the valuable time to read our manuscript and giving professional comments on our work. All the comments are helpful for improving the quality of our manuscript, which plays an important role in facilitating us to further understand the dynamic friction process and explain the mechanism. We have carefully considered these comments and have made detailed revisions as best as we could. Some important experimental and simulation results are supplemented as well as some statements are modified according to your suggestions. We hope the explanation and the revision can make the questions clear.

a) After the first cycle of friction test from 300K to 50K, the graphite nano-rollers have been already formed. Why does the friction coefficient become larger again in the following test at 300K?

Response:

Thank you for this comment. Although the graphite nano-rollers have been already formed after the first cycle of friction test from 300K to 50K, the graphite nano-rollers are unable to achieve the same lubrication performance as before in the following friction process. There are two main reasons. **Firstly**, this experiment adopts a ball-disc contact and rotational motion mode (Figure R1). At the contact interface between the ball and the disc, the shear force in situ drives the formation of the nano-rollers arranged

in order along the sliding direction at cryogenic temperature, which can play the role of rolling lubrication. However, the movement speed of the macroscale friction-paired ball is much faster than that of the nano-rollers, and thus the nano-rollers can not always follow and maintain at the ball-disk contact interface playing the lubricating role. **Secondly**, when the ball sliding once again on the original nano-roller, it has already deviated from the original position or moved out of the wear track under the action of rotational tangential force and motion inertia, and no longer effectively plays a lubricating role. Therefore, in this rotating-friction experiment, the formed nano-roller can only provide lubrication at once time, and it requires continuous supplementation. When the temperature increases to 300K, the nano-roller can not be formed continuously, so the friction coefficient becomes higher in the following test at 300K.

Figure R1 Schematic of friction experimental process.

b) It is still not so clear how the nano-rollers are formed in friction experimentally.

Response:

Thanks for your valuable comment. Figure R2 shows the experimental evolution of the nano-rollers formation in friction. We think it can be divided into three processes. The test sample is graphite block with multi-layered structure. **In the process 1**, at the initial friction stage, the friction force between the steel ball and graphite surface is higher than the van der Waals force between graphite interlayers, and thus graphite is delaminated and scratched to form graphite nanosheets with several layers (for example 2-3 layers in Figure R2). **In the process 2**, the delaminated graphite nanosheets cover on the surface of graphite blocks. The upper surface of the graphite nanosheets is

exposed to a vacuum environment of 300K, while the lower surface contacts with the graphite block at 50K. The temperature difference between the upper and lower surfaces of the graphite nanosheet drive the self-curling of the graphite nanosheet edges as demonstrate in the original manuscript. **In the process 3**, under the action of shear force, the self-curling graphite nanosheet is further curled to form intact and parallel nano-roller layer-by-layer. In this situation, the nano-rollers can be formed continuously during the cryogenic friction process. The corresponding SEM and TEM morphologies of experimental results are also attached to support the above structural evolution.

Figure R2 Schematic of the nano-rollers formation in friction process and experimental results of SEM and TEM morphologies.

c) The authors emphasize the important role of nano-rollers in reducing friction coefficient. How about simply introducing carbon nanotubes at the frictional interface? Could this produce a similar result?

Response:

Thanks for your comment. We have supplemented the corresponding friction

experiment by active introducing carbon nanotubes at the frictional interface. The tribological properties of the graphite nanosheets and graphite nanosheets/carbon nanotubes composite coatings are compared by the CSM tribometer (test parameters are shown in Figure R3). It can be seen that the addition of carbon nanotubes does not significantly affect the friction coefficient of graphite nanosheets (Figure R3a). From the SEM morphology of the wear tracks (Figure R3b), it can be observed that the randomly added carbon nanotubes cannot be arranged in order during the friction process and thus do not play roles in rolling lubrication. Therefore, as stated in the response to question a), it is important for in situ formation of the nano-roller induced by friction, which is arranged parallelly and along the sliding direction, can effectively play the role of rolling lubrication.

Figure R3 (a) Friction coefficient curves of the graphite nanosheets and graphite nanosheets/carbon nanotubes composite coatings. (b) SEM morphology of the wear track of graphite nanosheets/carbon nanotubes composite coating.

d) As the temperature decreases from 300K to 77K, self-curling of edges of graphite occurs. The curling is thought to be closely related to the formation of nano-rollers in the subsequent friction. Before friction tests, does the curling still exist when keeping temperature at 77K?

Response:

Thanks for your constructive comment, which inspired us to think more deeply. In fact, before friction tests, the curling still exists when keeping temperature at 77K. It is because when the temperature decreases from 300K to 77K, there is a temperature difference between the top and bottom surface of the graphite nanosheet, which would

produce stress owing to the uneven atomic shrinkage. If the stress is high enough, the self-curling deformation will occur, and the stress is released. When the temperature of the nanosheet reaches equilibrium and is maintained at a specific temperature (for example 77K), there is no other stress to drive it to restore the flat structure. This conclusion can be confirmed by the following experimental results. **In case 1**, during the in-situ cryogenic TEM characterization, the graphite nanosheets are induced to undergo self-curling as the temperature decreases from 300K to 77K (Figure R4a and b), and the microstructure of graphite nanosheets with edge-curling can keep at 77K for a long time without change. And as the temperature gradually returned to 300K, the microstructure of graphite nanosheets with edge-curling also can keep. **In case 2**, when the graphite nanosheets is immersed in liquid nitrogen and then taken to air at room temperature (about 300K) for TEM observation, the edge self-curling along the long-edge of the graphite nanosheets is still observed (Figure R4c and d).

Figure R4 Experimental process in case 1 and case 2, and the corresponding TEM images.

e) The authors accused the formation of nano-roller to be the internal stress induced by the temperature gradient. However, the stress is estimated to be only on the order of 10MPa (Fig. 3i). How does such a small stress cause an obvious curling at edges?

Response:

Thank you very much for pointing out this issue. Although, the stress is estimated to be about 10MPa under temperature gradient from 70K to 300K, the graphite nanosheet indeed rolled in MD simulations. Based on the following three facts, it should be able to be curled under a stress of 10MPa. **Firstly**, the adhesion between the graphite substrate and graphite nanosheets delaminated from graphite substrate is very weak and can be neglected. **Secondly**, the experimental shear strength of graphite nanosheet is about 60MPa-140MPa [PRL 108, 205503 (2012)], which inhibits the release of 10MPa stress directly through the lateral contraction of the low temperature layer. **Thirdly**, the required stress for bending deformation of graphite nanosheet with two or three layers is equivalent to the order of 10MPa. We prove it based on the assumption that the work done by the stress is greater than the energy to overcome the curling deformation during curling process. As shown in Figure R5, the graphite nanosheet is curled for one cycle under the force F caused by the temperature gradient field, which is equivalent to the top layer shrinking by $2\pi(n-1)d$, and the work done by the force is

$$W_F = F * 2\pi(n-1)d$$

where n is the number of layers, d is the interlayer spacing and $F = \tau * l * b$ holds. τ is the stress caused by the temperature gradient field. l and b is respectively the length and width of the graphite nanosheet. Therefore, it can be obtained that

$$W_F = \tau * l * b * 2\pi(n-1)d$$

Meanwhile the deformation energy of graphite nanosheet rolled for one cycle is

$$W_{\text{def}} = \frac{1}{2} D * \left(\frac{1}{R}\right)^2 * 2\pi R * b = D \frac{\pi b}{R}$$

where, R is bending radius and D is the bending stiffness, which relies significantly on the number of layers and has been carefully studied in [PRL 106, 255503 (2011)] for graphite nanosheet with no more than 5 layers. Here, by fitting with the power function, we obtained the D of the graphite nanosheet within 10 layers, as shown in Figure R5b.

To bend the graphite nanosheet, the work done by the force F caused by the

temperature gradient field should be greater than the deformation energy, that is $W_F \geq W_{\text{def}}$. We can obtain that

$$\tau \geq \frac{D}{2Rld} * \frac{1}{n-1}$$

Here, let's consider a typical graphite nanosheet with length of $2.46\mu\text{m}$. The bending radius is about 5nm . The values of l and R is from the MD simulations, which is consistent with the experimental result in magnitude. Base on it, we obtained the critical stress τ to roll graphite nanosheet for one cycle, as shown in Figure R5c. The stresses of graphite nanosheet with 2 and 3 layers are 3.1MPa and 6.1MPa , which are equivalent to the order of 10MPa caused by the temperature gradient. As the number of layers increases, however, the internal stress required becomes greater, making it less likely to curl only by the temperature gradient. We have added relevant expressions in the revised manuscript and supplementary information.

Figure R5 The critical stress to curl the graphite nanosheet. (a) Schematic diagram of the graphite nanosheet from plane to curling at edges. The length, width and thickness of the graphite nanosheet are l , b and $(n - 1)d$ respectively, where n is the number of layers and d is the interlayer spacing. (b) Bending stiffness curve of graphite nanosheet with number of layers. Short dotted line is the fitting of the experiment data with $D = a_1 * n^2 + a_3$. (c) The critical stress curve to curl the graphite nanosheet with the number of graphite nanosheet layers. The values of l and R is from the MD simulations, which is consistent with the experimental result in magnitude.

f) The graphite has a very low thermal conductivity along the out-of-plane direction. Even the rolling is induced by temperature gradient, the internal stress may have a large

difference in two or three layers. Only a small number of layers could be affected. Here the nano-rollers formed in experiments usually have a large thickness with the layer number more than 10. Could the authors provide more discussion on it?

Response:

Thank you very much for this insightful comment. Indeed, the multi-layer nanosheets (for example 10 layers) are difficult to form nano-rollers. Although the nano-rollers look like with the layer number of more than 10, it is not directly formed by rolling the graphite nanosheet with 10 layers, but it usually formed by rolling few-layer graphite nanosheets (1-3 layers) over multiple cycles layer-by-layer. This was verified by carefully examining the number of graphite nanosheet layers that formed the nano-rollers in the experiment and simulations.

As for the experiment, we have conducted extensive HRTEM testing and analysis. Due to the stacking of nano-rollers together, it is difficult to distinguish the number of layers for the rolling unit. Table R1 shows some examples that the rolling unit can be distinguished from the head or tail. It can be seen the rolling units are basically 1-3 layers, and the total layer number (white markings) of the ultimately formed nano-rollers is just the integer multiple of that of the rolling unit (blue markings), which indicates that the nano-rollers are formed via rolling of the few-layer graphite nanosheets (1-3 layers) from head to tail.

Table R1 Some HRTEM examples that the rolling unit can be distinguished.

As for the simulation, we further investigate the curling of 3-layer and 8-layer graphite nanosheets in the temperature gradient field from 300K-70K by MD simulations. As shown in Figure R6, the 3-layer graphite nanosheet curled, while there was no sign of curling in the 8-layer graphite nanosheet until the stress was completely released. As we discussed in the previous question of e), the internal stress to curl it becomes greater with increasing the number of graphite nanosheet layers. And the stresses of graphite nanosheet with 2 and 3 layers are 3.1MPa and 6.1MPa respectively, which are equivalent to the stress of 10MPa caused by the temperature gradient. Therefore, the temperature gradient should be the main reason for the curling at edges of graphite nanosheets. However, the internal stress to curl graphite nanosheet with 8 layers reaches 37.7MPa, which is difficult to curl it by the internal stress of 10MPa. In addition, the

low thermal conductivity makes it difficult for graphite nanosheet with 8 layers to form a uniform temperature gradient, which further hinders the process of temperature gradient driving it to curl.

The theoretical simulation results are consistent with the experimental results, and indicate that the nano-rollers are formed via rolling of the few-layer graphite nanosheets (1-3 layers) from head to tail. We have added the corresponding results and discussions in the revised manuscript and supplementary information.

Figure R6 The curling at edges of graphite nanosheets with different layers. (a) Models used in MD simulations. The final state and corresponding stress evolution of the curling at edges of 2-layer graphite nanosheet (b), 3-layer graphite nanosheet (c) and 8-layer graphite nanosheet (d). An unexpected result is that the stress on the top layer of the 8-layer graphite nanosheet reaches 30MPa, and the stress on the bottom layer is still about 10MPa. By careful analysis of the structure, it can be seen that there was abnormal elongation at the top layer, which did not completely contract until about 5ns, as shown in the insert in (d). This resulted in a residual stress of about 10MPa in the upper layer even when the lower layer stress was almost completely released. Therefore, it is a reasonable assumption that the effective stress is still about 10MPa in 8-layer graphite nanosheet.

g) It is proposed the possible contribution of phonons on the formation of nano-rollers. The authors further extend this to the so-called macroscale quantum effect. If so, this

may cause more comments on it. For example, how the coupling between phonon and electron on the roller formation?

Response:

We sincerely appreciate the professional comments. We have studied the related knowledge of quantum effect earnestly, and recognized your description is indeed correct. The core of macroscale quantum effects is the discontinuity of energy, involving phonons, electrons, acousoelectric coupling and other complex factors. In fact, our present work still belongs to the category of classical mechanics, which can only support the influence of atomic vibration on stress transfer. Therefore, we have removed the expressions such as “macroscale quantum effect” and “phonon” etc. in the revised manuscript. We have referred to some literatures [1-4], and corrected the expression as “the effect of atomic vibration on the formation of nano-rollers”. Thanks for your constructive suggestions, which help to improve the quality of our article and avoid ambiguity. We sincerely hope that the present expression is appropriate, and look forward to your valuable guidance kindly.

References

1. Y. Chen, F. Zhu, J. Leng, T. Ying, J. Jiang, Q. Zhou, T. Chang, W. Guo, H. Gao, Fluctuotaxis: nanoscale directional motion away from regions of fluctuation. *Proc. Natl. Acad. Sci. U. S. A.* **120** (31), e2220500120 (2023).
2. Allen, C. S. Liberti, E. Kim, J. S. Xu, Q. Fan, Y. He, K. Robertson, A. W. Zandbergen, H. W. Warner, J. H. & Kirkland, A. I. Temperature dependence of atomic vibrations in mono-layer graphene. *J. Appl. Phys.* **118**, 074302 (2015).
3. Yan, X. Distinguishing atomic vibrations near point defects. *Nat. Mater.* **22**, 540–541 (2023).
4. Lu, X. Feng, S. Li, L. Wang, L. Liu, R. Depicting defects in metallic glasses by atomic vibrational entropy. *J. Phys. Chem. Lett.* **573**, 247–250 (2019).

Reviewer #2 (Remarks to the Author):

Whether rolling friction of nanoparticles plays a role in decreasing friction has long been controversial in the area of tribology. Here by combining experimental and theoretical investigations, the article reported an interesting self-curling of graphite nanosheet at cryogenic temperature. Especially, the authors provide direct observation of the paralleled nanoroller structure on the frictional surface, and hence convince the important role of rolling friction in the system. On the other hand, it was generally accepted that friction of materials will increase with decreasing temperature due to thermal activation effect. However, the authors reported an unusual behavior for graphite nanosheet showing ultralow friction at cryogenic temperature, which is of potential use in industrial applications. I recommend acceptance of the article after some minor points are addressed.

Response:

We would like to express our appreciation for your constructive suggestions and valuable recommendations on our work. Thanks for your approbation of rolling lubrication research and the professionalism of your review comments, which gives a great improvement for our manuscript. We have carefully considered these questions and have made detailed revisions as best as we could. We hope the explanation and the revision can make the questions clear.

1. Full name for abbreviation GNS should be given. Page 3 Line 66, CNS should be GNS?

Response:

Thanks for your careful checks. We have rectified the abbreviation from CNS to GNS in the revised manuscript.

2. It seems that Figure 1c and Figure 1d were not ordered according to text.

Response:

Sorry for the disordered of Figure 1c and Figure 1d, we have made the corrected order in red in the revised manuscript and listed it below.

Fig. 1 | The illustration of the molecular rolling lubrication of graphite at cryogenic temperature. (a) Schematic of layer-layer sliding and molecular rolling lubrication as well as the structural evolution. (b) Microstructure of the friction interface at 300 K. (c) **Microstructure of the friction interface at 50 K.** (d) **The enlarged view of the formed nanoscroll.** (e and f) TEM images of the axial view of the formed graphite nanoroller. (g) SEM morphology of the friction interface at 50 K.

3. In molecular dynamics simulation, during the cooling of the cold side from 300 K to 70 K for 1 ns period, how does the temperature of the hot side change? Please clarify.

Response:

Thanks for your meaningful comments. We have added the temperature curve of the hot side (Figure 3i), which is an important data. The temperature of the bottom layer is fixed at 300 K during the cooling process of the top layer from 300 K to 70 K, and we also are added it into the revised manuscript in Figure 3.

Fig. 3i | Changes of stress of upper and lower layers of GNSs along x-direction with time (stage I corresponds to the temperature dropping of the atoms in the upper layer from 300 K to 70 K, and the temperature keeps constant at stage II).

4. The nanoroller structure (cylindrical shape) should be different from onion-like structure, which by name is of spherical shape. Please correct the use of onion-like in the text.

Response:

Thank you for your kind reminder. We have corrected the “onion-like” to “nanoscroll” in the revised manuscript.

“While at 50 K, a large amount of ordered **nanoscroll** crystal lattices are formed at the friction interface (Figure 1c, Figure S1 and S6). The enlarged view (Figure 1d) of the **nanoscroll** suggests that it is formed via rolling of trilayer GNSs from head to tail, and the lattice spacing (0.34 nm) is consistent with the *d*-spacing of the (002) basal plane of graphite. Corresponding axial views (Figure 1e and 1f) of the **nanoscroll** lattices reveal that they are nanoroller structure with the length of hundreds of nanometers, and the lattice spacing is consistent with (100) crystal plane spacing of graphite.”

5. Page 8 Line 170, 4f should be 4e.

Response:

Thanks for your correction. We have corrected 4f to 4e in the revised manuscript.

“Simulations indicate that the distribution of atomic stress is dispersed at 300 K, while it becomes concentrated at 70 K (Figure 4d and 4e).”

REVIEWERS' COMMENTS

Reviewer #1 (Remarks to the Author):

Dear Editor,

The authors have addressed all my comments. I now recommend it for publication.

all the best
Suzhi Li

Reviewer #2 (Remarks to the Author):

The authors have addressed the questions properly. The manuscript is ready to publish.